# From Question to Exploration: Can Classic Test-Time Adaptation Strategies Be Effectively Applied in Semantic Segmentation?

## ABSTRACT

Test-time adaptation (TTA) aims to adapt a model, initially trained on training data, to test data with potential distribution shifts. Most existing TTA methods focus on classification problems. The pronounced success of classification might lead numerous newcomers and engineers to assume that classic TTA techniques can be directly applied to the more challenging task of semantic segmentation. However, this belief is still an open question. In this paper, we investigate the applicability of existing classic TTA strategies in semantic segmentation. Our comprehensive results have led to three key observations. First, the classic normalization updating strategy only brings slight performance improvement, and in some cases it might even adversely affect the results. Even with the application of advanced distribution estimation techniques like batch renormalization, the problem remains unresolved. Second, although the teacher-student scheme does enhance the training stability for segmentation TTA in the presence of noisy pseudo-labels and temporal correlation, it cannot directly result in performance improvement compared to the original model without TTA under complex data distribution. Third, segmentation TTA suffers a severe long-tailed class-imbalance problem, which is substantially more complex than that in TTA for classification. This long-tailed challenge negatively affects segmentation TTA performance, even when the accuracy of pseudo-labels is high. Besides those observations, we find that visual prompt tuning (VisPT) is promising in segmentation **TTA**. Further, we propose a novel benchmark named **TTAP** based the above findings and Vis**P**T. The outstanding performance of TTAP has also been verified. We hope the community can give more attention to this challenging, yet important, segmentation TTA task in the future. The source code will be publicly available.

## CCS CONCEPTS

• **Computing methodologies** → **Learning under covariate shift**.

## KEYWORDS

Test-time adaptation; semantic segmentation; vision transformer

## 1 INTRODUCTION

Test-time adaptation (TTA) focuses on tailoring a pre-trained model to better align with unlabeled test data at test time [36]. That model

Permission to make digital or hard copies of all or part of this work for personal or classroom use is granted without fee provided that copies are not made or distributed for profit or commercial advantage and that copies bear this notice and the full citation on the first page. Copyrights for components of this work owned by others than the author(s) must be honored. Abstracting with credit is permitted. To copy otherwise, or republish, to post on servers or to redistribute to lists, requires prior specific permission and/or a fee. Request permissions from permissions@acm.org.

*ACM MM, 2024, Melbourne, Australia*

© 2024 Copyright held by the owner/author(s). Publication rights licensed to ACM.
ACM ISBN 978-x-xxxx-xxxx-x/YY/MM
https://doi.org/10.1145/nnnnnnn.nnnnnnn

needs to simultaneously produce a prediction and adapt itself in the online manner. The TTA paradigm is popular since the test data may unavoidably encounter corruptions or variations, such as Gaussian noise, weather changes, and many other reasons [9, 19]. Furthermore, the training and test data can not co-exist due to privacy concerns. These challenges have propelled TTA to the forefront as an emergent and swiftly evolving paradigm [22, 29, 30, 36, 40]. Broadly, existing techniques can be classified into two main categories: Test-Time Training (TTT) [25, 36] and fully TTA [29, 40]. Compared to TTT, fully TTA (TTA for short) is more practical and it is also the focus of this paper, since TTT needs to change the original model training which may be infeasible due to privacy concerns.

The key idea of TTA methods is to define a proxy objective at test time to adapt the pre-trained model in an unsupervised manner. Typical proxy objectives include entropy minimization [40], pseudo labeling [23] and class prototypes [35]. While the majority of TTA studies have centered on classification problems, real-world scenarios frequently highlight the ubiquity and critical nature of semantic segmentation. A prime instance is autonomous driving, where each system must accurately and instantaneously segment an array of dynamic and unpredictable perceptions [20]. A segmentation task is much more challenging than an image-level classification counterpart. For example, it is extremely difficult to estimate pixel-level data distribution which may result in error accumulation, the long-tailed (LT) problem brings serious class imbalance, low-quality pseudo-labels of pixels may cause model collapse, etc. Numerous newcomers and engineers might mistakenly believe that classic TTA techniques can be directly applied in semantic segmentation. Nevertheless, this assumption still remains unverified, posing an open question. Thus, the TTA community needs to answer this open question: Can classic test-time adaptation strategies be effectively applied in semantic segmentation?

In this paper, we attempt to address this question and provide systematic studies to assist both experienced researchers and newcomers in better understanding segmentation TTA. To the best of our knowledge, this paper is among the first to comprehensively investigate classic TTA techniques for semantic segmentation. Our main observations are summarized as follows:

- Normalization statistics are frequently used in classification TTA [29, 30, 40]. However, we find that the classic normalization updating strategy offers marginal performance gains and can sometimes even deteriorate the outcomes of segmentation TTA. Advanced techniques like batch renormalization and large batch sizes fail to address this limitation effectively. This observation motivates us to consider the update of other modules to estimate the data distribution. We find that updating the attention module in Transformer [56] can promote the performance in segmentation TTA.

- While the teacher-student (TS) scheme bolsters training stability in segmentation TTA amidst noisy pseudo-labels and different orders of images, we find that it does not always elevate the performance beyond models not employing TTA, especially in scenarios involving complex data distribution (i.e., continual TTA) [42]. Instead, we find that the TS scheme can produce high-quality pseudo-labels in segmentation TTA, compared to the single-model.
- Segmentation TTA grapples with an acute LT imbalance issue, which is more intricate than its counterpart in classification TTA. We find that this LT dilemma profoundly impedes segmentation TTA efficacy, even with high-accuracy pseudo-labels. Instead, we discover that the introduction of region-level solution can improve the performance in segmentation TTA.

In light of the above observations and comparisons, we discover that visual prompt tuning (VisPT) is a promising solution in segmentation **TTA**. Moreover, we find that combining RGB and frequency domain can uncover a richer set of image priors, which is valuable for the creation of visual prompts. Based on Vis**P**T and the findings, we propose a novel benchmark named **TTAP** which has been verified to be effective in segmentation TTA. In particular, its computational time is much less than the comparative approaches.

## 2 RELATED STUDIES

*Classic test-time adaptation.* Normalization statistics are widely used in TTA to compute the data distribution based on the test data. TENT [40] adapts batch normalization (BN) layers based on entropy minimization, i.e., the confidence of the target model is measured by the entropy of its predictions. EATA [29] actively selects reliable samples to minimize entropy loss during inference. Furthermore, it also introduces a Fisher regularizer to filter out redundant samples to reduce the computational time. SAR [30] is a reliable and sharpness-aware entropy minimization approach that can suppress the effect of noisy test samples with large gradients. ATP [2] is flexible to handle various kinds of distribution shifts in online federated learning, by adaptively learning the adaptation rates for each target model. However, the cross-entropy loss, which is effectively used in classification, is inherently inapplicable to a regression problem such as pose estimation [21].

Besides entropy-based approaches, many other strategies are also introduced to address TTA. TEA [49] transforms the source model into an energy-based classifier to align the distributions of the model and test data. AdaContrast [3] combines contrastive learning and pseudo labeling to handle TTA. AdaNPC [52] is a parameter-free TTA approach based on a K-Nearest Neighbor (KNN) classifier, where the voting mechanism is used to attach labels based on $k$ nearest samples from the memory. Different from traditional approaches, CTTA-VDP [6] introduces a homeostasis-based prompt adaptation strategy that freezes the source model parameters during the continual TTA process. Based on a large-scale open-sourced benchmark approaches and thorough analysis, TTAB [55] unveils three pitfalls in prior TTA approaches under classification tasks.

*Semantic segmentation.* Pixel-level annotation is one of the key characteristics of semantic segmentation. HAMLET [4] can handle unforeseen continuous domain changes, since it combines a specialized domain-shift detector and a hardware-aware backpropagation orchestrator to actively control the model's real-time adaptation for semantic segmentation. CoTTA [42] can reduce error accumulation based on weight-averaged and augmentation-averaged predictions. Segmentation tasks are also pervasive in medical images, since the scanner model and the protocol differ across different hospitals. This issue can be handled by introducing an adaptable per-image normalization module and denoising autoencoders to incentivize plausible segmentation predictions [16].

SITA [17] can be applied in segmentation and the source model is adapted independently based on each individual test sample which will be augmented several times. DIGA [43] is a backward-free segmentation approach that is based on a semantic and a distribution adaptation module, which can adapt the model at both semantic and distribution levels. However, the weights of different modules are fixed. Segmentation TTA has also been extended to multi-modal 3D tasks based on intra-modal pseudo-label generation and inter-modal pseudo-label refinement [34], although the experiments are carried out on simple scenarios. OASIS [39] is a training-validation-deploy benchmark that focuses on the evaluation protocol, adaptation benchmark and impact of catastrophic forgetting.

Similar to TTAB [55], the segmentation TTA community also lacks insightful guidelines. For instance, are classic TTA strategies, such as normalization and teacher-student (TS) scheme still effective in segmentation TTA? What is the challenge to address LT problems? Are classic TTA techniques robust to batch dependency of the test data? What kind of deep architecture is preferred, Transformer or CNN [56]? Moreover, what are the possible solutions to improve segmentation TTA when classic strategies fail to work?

## 3 PRELIMINARIES

### 3.1 Problem Statement

Let $\mathcal{D}^{train} = \{(\mathbf{x}_i, \mathbf{y}_i)\}_{i=1}^{N} \in \mathcal{P}^{train}$ be the training data, where $\mathbf{x}$, $\mathbf{y}$ and $N$ represent the features, labels and data amount, respectively. Let $f_{\Theta}(\mathbf{x})$ denote a pre-trained segmentation model with parameters $\Theta$. The goal of segmentation TTA is to adapt $f_{\Theta}(\mathbf{x})$ to the unlabeled test data $\mathcal{D}^{test} = \{\mathbf{x}_i\}_{i=1}^{M} \in \mathcal{P}^{test}$ with different data distribution, i.e., $\mathcal{P}^{train}(\mathbf{x}) \neq \mathcal{P}^{test}(\mathbf{x})$. Under the TTA paradigm [40], the model $f_{\Theta}(\mathbf{x})$ receives a batch of unlabeled test data at each time step, and it will be updated in an online manner.

### 3.2 Classic TTA Strategies

In this paper, our primary objective is to uncover the unique challenges posed by segmentation TTA under classic strategies and provide some inspirational solutions. To achieve that purpose, we delve into several well-established strategies, including normalization updating [54], teacher-student (TS) scheme [42], test-time augmentation (Aug) [26], and pseudo labeling (PL) [52], all of which have demonstrated their effectiveness in classification TTA.

### 3.3 Experimental Setups

To ensure consistent evaluations of various TTA approaches, we conduct empirical studies based on several widely used semantic

**Table 1: Results of batch norm updating strategies (i.e., TENT [40] and its variants) on datasets ACDC, Cityscapes-fog and Cityscapes-rain (mIoU, %). SO indicates using the source model without adaptation, while BS represents the batch size of test data at each iteration. Except that the TENT (larger BS) variant uses a batch size of 4, the other methods are based on BS = 1 as mentioned in Section 3.**

| Method | A-fog | A-night | A-rain | A-snow | CS-fog | CS-rain | Avg. |
|---|---|---|---|---|---|---|---|
| SO | 68.2 | 39.5 | 59.7 | 57.6 | 74.2 | 66.6 | 61.0 |
| TENT | 63.3 (-4.9) | 39.5 (-0.3) | 57.6 (-2.1) | 54.9 (-2.7) | 73.9 (-0.3) | 66.8 (+0.2) | 58.8 (-2.2) |
| TENT (larger BS) | 64.4 (-3.8) | 39.8 (+0.3) | 57.3 (-2.4) | 54.0 (-3.6) | 71.6 (-2.6) | 66.7 (+0.1) | 59.0 (-2.0) |
| TENT (BN-fixed) | 68.1 (-0 1) | 39.4 (-0.1) | 60.1 (+0.4) | 57.1 (-0.5) | 74.1 (-0.1) | 66.5 (-0.1) | 59.9 (-0.1) |
| BN adapt | 62.0 (-6.2) | 37.3 (-2.2) | 55.1 (-4.6) | 52.7 (-4.9) | 73.3 (-0.9) | 65.9 (-0.7) | 57.7 (-3.3) |
| AugBN | 67.6 (-0.6) | 38.2 (-1.3) | 59.0 (-0.7) | 56.3 (-1.3) | 73.3 (-0.9) | 65.9 (-0.7) | 60.0 (-1.0) |

segmentation datasets, including ACDC [32], Cityscapes-foggy (CS-fog) [31] and Cityscapes-rainy (CS-rain) [13]. In addition, we strictly follow the implementation details outlined in previous studies [4, 42], and use Segformer-B5 [46] as the pre-trained model. Two state-of-the-art and recent segmentation approaches, i.e., Oneformer [14] and SAM [18], are also used in comparative experiments. We focus on transformer-based architectures instead of CNN-based architectures, since the former exhibits more promising results than the latter (cf. Appendix 1). Unless otherwise specified, all experiments are conducted with a batch size (BS) of 1, mirroring real-world scenarios where the test samples often arrive one by one in an online manner. Some of the experimental results, i.e., Tables and Figures, are displayed in the Appendix. The choice of hyper-parameters can be seen in the code of this paper which will be publicly available.

## 4 DOES NORMALIZATION UPDATING WORK FOR SEGMENTATION TTA?

### 4.1 Norm Updating Fails in Segmentation

We start with batch normalization (BN) updating strategies [28, 33]. Most existing BN-based TTA methods [29, 40], contrary to typical deep learning pipelines, compute the distribution statistics directly from the test data, rather than starting with or inheriting those from the training data. These methods only update the BN layers during TTA, restricting changes exclusively to the model parameters. This ensures that the core learned features remain intact, while only the normalization gets adjusted based on the test data. These approaches have demonstrated their effectiveness in bridging domain gaps for image classification at test time, however, their efficacy in semantic segmentation is yet to be thoroughly explored and validated.

To delve deeper into this, we conduct a thorough evaluation of BN-based TTA methods in segmentation based on a classic method TENT [40]. Specifically, TENT adapts a model by using the BN statistics from mini-batch test data (with BS = 1) instead of those inherited from the training data, and updating the affine parameters of BS through entropy minimization. Moreover, we explore two variants of TENT: 1) TENT (larger BS) seeks to enhance TENT's performance by utilizing a larger batch size of 4, aiming for a more precise estimation of distribution statistics; 2) TENT (BN-fixed) retains the BN statistics from the training data without adaptation and solely updates the affine parameters of BS through entropy minimization. Finally, we also conduct comparisons with BN adapt [33]

and AugBN [17], both of which have demonstrated their effectiveness in segmentation TTA using CNN-based architectures [17].

As shown in Table 1, we have three main observations. First, all TENT variants perform worse than the *Source Only* (SO), highlighting the difficulties that classic batch norm updating methods encounter in segmentation TTA. Second, even though using a larger batch size marginally elevates TENT's performance, it remains overshadowed by SO. Last, the TENT (BN-fixed) variant achieves performance only similar to SO, although the affine parameters of BN are updated. This shows that retaining the BN statistics from the training data plays a key role, while updating the affine parameters of BN does not bring the expected improvement. In summary, batch norm updating strategies, despite performing well in classification TTA, do not meet anticipated outcomes in segmentation TTA. Please refer to Section 4.3 for more discussions on distribution estimation tricks like larger batch size and batch renormalization.

### 4.2 Aligning Batch Norm Statistics Loses Its Magic in Segmentation

We next aim to probe the underlying reasons for the poor performance of BN-based TTA methods in semantic segmentation. Before diving into the detailed analysis, we first provide a foundational overview of BN updating to ensure clarity and comprehension. Let $f \in \mathbb{R}^{B \times C \times H' \times W'}$ represent a mini-batch of features, where $C$ indicates channel numbers, $H'$ is the height of features, and $W'$ is the width. BN normalizes $f$ using the distribution statistics of mean $\mu$ and variance $\sigma$ (both $\mu$ and $\sigma$ belong to $\mathbb{R}^C$). The normalization is mathematically expressed as:

$$f^* = \gamma \cdot f' + \beta, \quad where \quad f' = \frac{f - \mu}{\sigma}, \tag{1}$$

where $\gamma, \beta \in \mathbb{R}^C$ are learnable affine parameters of BN that represent scale and shift, respectively. During inference, $\mu$ and $\sigma$ are set to $\mu^{ema}$ and $\sigma^{ema}$, respectively, which are the exponential-moving-average (EMA) estimation of distribution statistics. Previous BN-based TTA methods for classification have shown that in situations where there is a distribution shift between the training and test data, i.e., $\mathcal{P}^{train}(\mathbf{x}) \neq \mathcal{P}^{test}(\mathbf{x})$, replacing the EMA estimation of $\mu^{ema}$ and $\sigma^{ema}$ with the test mini-batch statistics can boost model performance [40] when test mini-batch statistics are accurate.

However, Table 1 has demonstrated that such a strategy does not make sense in semantic segmentation. The challenges arise from the model's difficulty in accurately assessing the test data statistics during adaptation for segmentation. To shed light on this, we visualize the estimated distribution statistics of BN in Figure 1 (a)-(b).

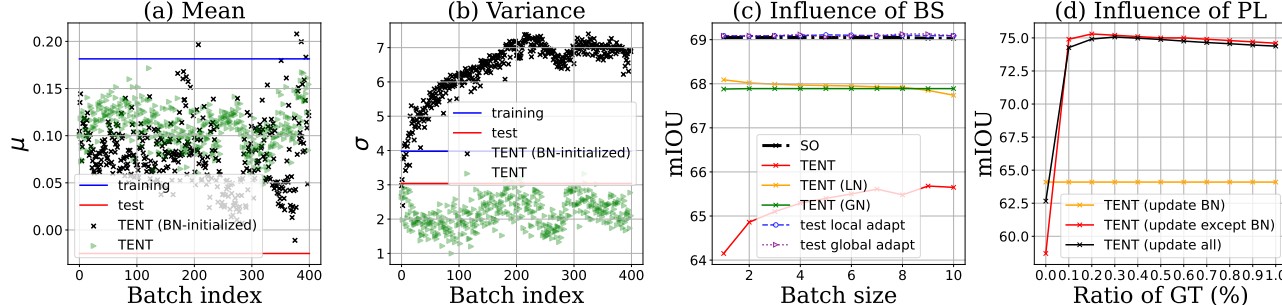

**Figure 1: Quantitative metrics analysis. (a) and (b) capture the BN distribution statistics through online adaptation. (c) shows the differential impacts of different batch norm updating techniques across different batch sizes (BS). (d) delves into the effects of varying updating strategies based on TENT, contrasting different proportions of pseudo-labels with the rest being ground-truth (GT) labels.**

To be specific, we train the model from scratch on both Cityscapes training data and ACDC-fog test data, followed by recording BN distribution statistics, represented by "training" (the blue line) and "test" (the red line) in Figure 1 (a)-(b). Subsequently, we employ the aforementioned TENT to adapt the trained model to test data, and record the change of BN distribution statistics. Specifically, TENT adjusts BS statistics based solely on mini-batch test data independently at each iteration. In contrast, TENT (BN-initialized) starts with the BN distribution statistics from the training data model and progressively adapts BN statistics using EMA, instead of computing statistics independently for each test batch.

Figure 1 (a)-(b) leads to four main findings. First, the distributional discrepancy between the "training" and "test" data is pronounced. Second, while TENT (BN-initialized) — represented by the black dots in Figure 1 (a)-(b) — does endeavor to adjust to the test data, it fails to estimate the test data very well, still remaining misalignment relative to the true test data distribution. Third, the BN statistics' evolution in TENT (depicted by the green points) mirrors that of TENT (BN-initialized) closely. This resemblance arises because, even though TENT's BN statistics are not inherited and are recalibrated based on individual mini-batches of test data at every iteration, the rest of the model parameters are indeed derived from the training data model. Consequently, the initial feature distribution still aligns more closely with the training data's distributional characteristics, preventing direct approximation of the test data distribution. As the adaptation progresses, while there is a trend towards aligning with the test distribution, it, much like TENT (BN-initialized), ultimately fails to capture that distribution accurately. Last, we notice a pronounced increase in the variance of TENT (BN-initialized), indicating a widening divergence in the distribution estimation. In summary, the imprecise estimation of the test data distribution renders BN updating ineffective for segmentation TTA, with the fluctuating and escalating variance even potentially imparting detrimental effects on model performance.

## 4.3 Distribution Estimation Tricks Cannot Resolve the Problem

In light of the above discussions, we next ask whether further using distribution estimation tricks can rectify the issues associated with the distribution estimation of normalization updating in segmentation TTA. In response, we investigate three policies: harnessing a

larger batch size, adopting batch renormalization, and leveraging GT labels (mainly for empirical analysis).

*Larger batch size.* Previous studies [30, 40] have shown that using a larger batch size can enhance the BN updating for classification TTA. Driven by this rationale, we investigate the impact of different batch sizes (ranging from 1 to 10) on segmentation TTA, where we also provide the results based on layer normalization (LN) [1] and group normalization (GN) [45], which replace the BN to LN and GN, respectively. As shown in Figure 1 (c), an increase in batch size does indeed enhance BN updating. However, this enhancement does not translate to an improvement over SO, i.e., using the pre-trained source model without adaptation. This indicates that merely increasing the batch size cannot adequately solve the issue of normalization-based segmentation TTA methods. Furthermore, we also observe that the outcomes of GN are similar to LN, suggesting that the significance of normalization layers might not be so important as we previously expected.

*Batch renormalization.* Utilizing local test mini-batch statistics for model adaptation proves unreliable, especially when confronting persistent distribution shifts [48]. Such unreliability originates from error gradients and imprecise estimations of test data statistics. In response, we delve into two test-time batch renormalization techniques [48, 54], namely *Test Local Adapt* and *Test Global Adapt*, aiming to refine the distribution estimation. *Test Local Adapt* leverages the source statistics to recalibrate the mini-batch test data distribution estimation, whereas *Test Global Adapt* uses test-time moving averages to recalibrate the overall test distribution estimation. As shown in Figure 1 (c), while batch renormalization strategies do enhance the performance of TENT, their performance is just comparable to that of SO and cannot lead to performance improvement in semantic segmentation.

*Ground-truth labels.* To analyze the impact of pseudo-label noise on distribution estimation, we leverage true labels for empirical studies. Moreover, to analyze the effects of updating different network components, we further explore three distinct updating strategies. (1) TNET (update BN): the affine parameters in BN are updated; (2) TNET (update except BN): the parameters except for BN are updated; (3) TNET (update all): all the model parameters are updated. As shown in Figure 1(d), when solely relying on pseudo-labels,

Table 2: Results of the teacher-student scheme on ACDC (mIoU, %). "SO"/"Single"/"TS" are short for source only/the single-model/the teacher-student scheme, and "PL"/"Aug" are short for pseudo-labeling/test-time augmentation, respectively.

| Method | PL | Aug | A-fog | A-night | A-rain | A-snow | Avg. |
|--------|----|-----|-------|---------|--------|--------|------|
| SO | | | 68.2 | 39.5 | 59.7 | 57.6 | 56.3 |
| Single | ✓ | | 54.6 (-13.6) | 29.0 (-10.5) | 45.5 (-14.2) | 41.2 (-16.4) | 42.7 (-13.7) |
| TS | ✓ | | 67.4 (-0.8) | 38.7 (-0.8) | 59.8 (+0.1) | 57.2 (-0.4) | 55.9 (-0.4) |
| Single | ✓ | ✓ | 41.9 (-26.3) | 18.1 (-21.4) | 20.7 (-39.0) | 16.4 (-41.2) | 24.4 (-31.9) |
| TS | ✓ | ✓ | 70.0 (+1.8) | 40.2 (+0.7) | 63.8 (+4.1) | 59.2 (+1.6) | 58.4 (+2.1) |

Table 3: Comparisons between TENT [40] and its attention-based version (Attn) (mIoU, %). The results indicate that incorporating the attention mechanism can enhance the performance in TTA.

| Method | A-fog | A-night | A-rain | A-snow | CS-fog | CS-rain | Avg. |
|--------|-------|---------|--------|--------|--------|---------|------|
| TENT | 63.3 | 36.5 | 56.2 | 54.0 | 73.8 | 66.8 | 58.4 |
| TENT (Attn) | **69.2** | **39.1** | **61.2** | **58.3** | **74.1** | **67.2** | **61.5** |

TENT (update BN) outperforms its counterparts due to its minimal parameter updating, making it less susceptible to the noise of pseudo-labels. In contrast, the other baselines exhibit markedly inferior performance under these conditions. However, as the quality of pseudo-labels improves—with the incorporation of more GT labels, there's a significant performance boost in TENT (update except BN) and TENT (update all). Yet, TENT (update BN) remains stagnant, not showing the same enhancement. This further demonstrates the limitations of existing BN updating TTA strategies in semantic segmentation. Thus, what is the promising solution when distribution estimation tricks fail to work?

## 4.4 Updating the Attention Module is Promising

Based on the above analysis, we believe that: 1) it is hard to estimate the normalization statistics in segmentation TTA at the pixel-level[1]; 2) within the Transformer-based architectures, the impact of normalization layers is relatively muted compared to that in CNN-based architectures [30]. Thus, which module is important to estimate the data distribution in segmentation TTA?

We hypothesize that the self-attention mechanism may play a pivotal role in Transformer-based architectures [12]. This hypothesis is exemplified by analyzing Segformer-B5 [46], which utilizes a gradient-based sorting technique to arrange all layers, placing some attention modules and multi-layer perceptions (MLPs) ahead of the normalization layers. As displayed in Table 3, it indicates that updating the attention mechanism is a promising and novel direction for transformer-based models. In the future, focusing on the attention mechanism and the fusion of MLP modules may enhance the effectiveness of Transformer-based architectures in segmentation TTA.

## 5 DOES THE TEACHER-STUDENT SCHEME WORK FOR SEGMENTATION TTA?

## 5.1 The Teacher-student Scheme Helps Stabilize Segmentation TTA

The teacher-student exponential moving average (TS-EMA) scheme [10] has been shown to enhance model training and accuracy [37].

[1]We will discuss the region-level solution in Section 6.2

Table 4: Comparisons under different temporal orders of images on Cityscapes-fog and Cityscapes-rain (mIoU, %). Different random seeds (i.e., 0/9/99/999/999) represent different time orders.

| Domain | Single (GT) | TS | 0 | 9 | 99 | 999 | 9999 |
|--------|-------------|-----|------|------|------|------|------|
| CS-fog | ✓ | | 78.2 | 78.1 | 78.2 | 78.2 | 78.3 |
| CS-fog | | ✓ | 76.7 | 81.1 | 82.0 | 82.1 | 81.9 |
| CS-rain | ✓ | | 72.0 | 78.2 | 71.9 | 71.9 | 71.9 |
| CS-rain | | ✓ | 83.9 | 79.3 | 79.4 | 80.3 | 79.5 |

Many recent methods [38, 42, 48] introduce it into TTA by using a weighted-average teacher model to improve predictions. The underlying belief is that the mean teacher's predictions are better than those from standard and single models. However, the precise influence of TS-EMA on segmentation TTA has not been thoroughly investigated. In this Section, we seek to delve into its empirical impact. For the implementation of the TS-EMA scheme, we follow CoTTA [42] to update the student model by $\mathcal{L}_{PL}(\mathbf{x}_{\mathcal{T}}) = -\frac{1}{C}\sum_c \tilde{\mathbf{y}}_c \log \hat{\mathbf{y}}_c$, where $\tilde{y}_c$ is the probability of class $c$ in the teacher model's soft pseudo-labels prediction, $\hat{y}_c$ is the output of the student model, and $C$ indicates the total number of categories.

To figure out whether the TS-EMA scheme indeed stabilizes TTA for semantic segmentation, we compare the TS-EMA scheme and the single-model (Single) scheme with pseudo-labeling (PL) and test-time augmentation (Aug) [26]. As shown in Table 2, the Single scheme consistently underperforms compared to the SO baseline, a trend that persists even with the integration of PL and Aug. In stark contrast, the TS-EMA scheme maintains relatively stable performance. Using PL, it experiences only minor drops in categories like "A-fog" and "A-night", and even shows an improvement in "A-rain". Moreover, when employing both PL and Aug, TS outperforms the SO baseline. In light of these observations, we conclude that TS-EMA stands out as a robust method to improve the training stability of segmentation TTA.

*Temporal correlations.* Additionally, we also investigate the performance regarding the temporal order of samples. This consideration is practical since a TTA task should process each test instance online and independently. Comparing the TS scheme and the single-model (GT labels are introduced for further examination, since the pseudo-labels are found to contain serious noise in the single-model), the results are displayed in Table 4. Even with varying random seeds (i.e., time orders), the TS scheme consistently yields similar results, indicating that it is not susceptible to fluctuations in temporal correlations. In contrast, the results of the single-model exhibit more noticeable variations. For instance, when the seed is

set to 9, the result for CS-rain is 78.2%, whereas the results for other seeds hover around 72%.

## 5.2 Discussions of Potential Limitations

While previous analysis attests to the efficacy of the TS-EMA scheme, a closer examination of Table 2 (cf. Appendix) underscores a notable observation: when the SO baseline is fortified with test-time augmentation, its performance surpasses that of TS combined with both PL and Aug. This suggests that the primary advantage of TS-EMA may lie in mitigating the noise introduced by PL, thereby allowing Aug to function more effectively.

This finding provokes a subsequent question: if the accuracy of pseudo-labels is enhanced, would the TS model also exhibit improved performance as shown in previous studies [37]? To answer this question, we adjust the experimental setting, concentrating on situations where pseudo-labels become increasingly accurate, marked by a growing proportion of GT labels. In this context, we assume that the GT labels are accessible so that we can empirically assess the model performance across varying ratios of GT labels.

We continue to compare the single-model and the TS scheme. As depicted in Figure 1 (cf. Appendix), we have plotted the IoU (Intersection over Union) metrics for each class against varying levels of GT. This visualization helps us critically assess how the performance trajectory of these two schemes adjusts as the accuracy of the pseudo-labels promotes. For the sake of fair comparison, the policy of Aug is not adopted in that Figure, where comparative results indicate that the performance improvement will be minimal without data augmentation. This experiment aims to investigate the importance of each module of the TS scheme and emphasize the necessity of Aug in this scheme. Moreover, we also report the result of TS scheme leveraging data augmentation in Figure 2 (cf. Appendix).

Upon a detailed observation, it becomes evident that both the single-model and TS scheme exhibit similar performance trends. When the precision of the pseudo-labels hits an approximate threshold of 1%[2], the single-model scheme achieves a performance that is almost neck-and-neck with that of the TS scheme. However, as we progress beyond this pseudo-labels precision threshold, an interesting divergence arises: while the single-model continues to better its performance, the TS model appears to stagnate and its mIoU (mean IoU) metric remains static at 0.69. In stark contrast, the single-model exhibits a commendable improvement, witnessing its mIoU metric jump from an initial 0.59 to a robust 0.74.

Given this observation, one could infer a potential limitation intrinsic to the TS scheme. Despite having increasingly accurate pseudo-labels at its disposal, it does not exhibit the expected adaptability and responsiveness, unlike its single-model counterpart.

*Continual TTA.* Real-world perception systems operate in nonstationary and constantly evolving environments, where the test data distribution can change from time to time [42]. As shown in Figure 2, we sequentially adapt the pre-trained model of the dataset Cityscapes to the dataset ACDC. Surprisingly, the performance of the TS scheme gradually deteriorates and is comparable to that of TENT. In the end, the TS scheme even exhibits inferior performance

---

[2]To put this into perspective, for an ACDC image, 1% GT translates to a total of $0.01 * 1080 * 1920 = 22572$ pixels.

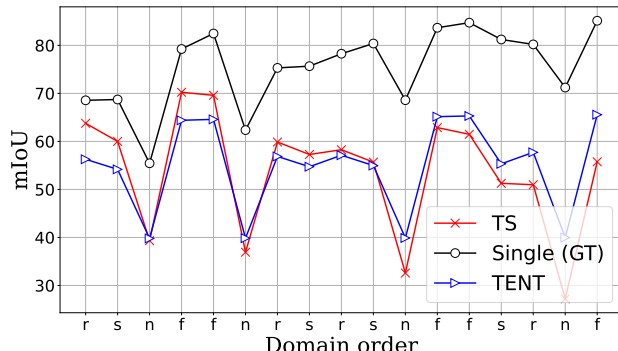

**Figure 2: The results of online continual segmentation TTA on the Cityscapes-to-ACDC task (%). We evaluate the four test conditions continually four times to evaluate the performance of long-term adaptation. "f"/"n"/"r"/"s" are short for domain fog/night/rain/snow, respectively.**

compared to TENT. In addition, we also use Single (GT) for examination. The results obtained with Single (GT) demonstrate that high-quality pseudo-labels can prevent the deterioration caused by the changing test data distributions.

Based on the above analysis, it is clear that the TS scheme is capable of achieving stable training, even in the presence of noisy labels or temporal correlation in TTA. However, we identify some challenges associated with the TS scheme: 1) it is difficult to effectively utilize high-quality pseudo-labels; 2) it tends to deteriorate under continual TTA. These findings highlight the need of further research and improvements to fully harness the potential of the TS scheme.

## 6 DOES CLASS IMBALANCE INFLUENCE SEGMENTATION TTA?

### 6.1 Segmentation TTA Suffers the Long-tailed Problem

Semantic segmentation inherently grapples with the challenge posed by data imbalance [11, 51]. Certain semantic classes, such as sky and buildings, are predisposed to occupy vast areas populated with significantly more pixels, often leading them to dominate the visual space, prevalent in numerous realistic pixel-level classification endeavors.

When placed in the context of TTA, the long-tailed (LT) problem becomes more pronounced, manifesting as an obvious bias in test-time optimization towards dominant classes [50, 54]. Both NOTE [8] and SAR [30] can handle class imbalance in classification TTA, however, they perform poorly when addressing the LT problem in segmentation TTA. As shown in Figure 5 (cf. Appendix), the numerical disparity between the majority and minority classes surpasses a staggering 1000-fold difference. This stark contrast is evident when compared to common datasets used in classification tasks, such as CIFAR10-LT, where the most majority class is only in the thousand-level range and has 100× more samples than the most minority class [44]. Adding to the challenge is the nature of semantic segmentation itself, which involves copious pixel-level labels, further complicating the LT complexity. In this Section, we

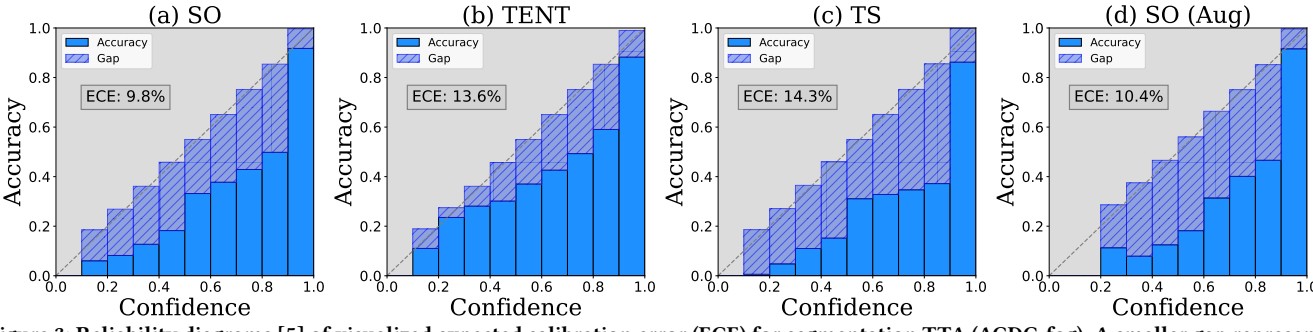

**Figure 3: Reliability diagrams [5] of visualized expected calibration error (ECE) for segmentation TTA (ACDC-fog). A smaller gap represents less ECE and better calibration. After adaptation, ECE actually becomes larger, indicating that the model is more over-confident.**

aim to shed light on the challenges of the LT problem as it manifests in segmentation TTA.

We then show the intricate complexity and challenge inherent in semantic segmentation, making it markedly more difficult than classification tasks. To delve deeper into this issue, we assume that the model can generate high-confidence pseudo-labels for the test data during adaptation and subsequently analyze the resultant state of the model. Our analysis will be conducted from three perspectives: examining the confusion matrix, conducting recall-precision analysis, and evaluating model calibration.

*Confusion matrix.* The confusion matrix of ACDC-fog is displayed in Figure 9 (cf. Appendix), unveiling extreme variations in the outcomes for each class, reflecting the substantial discrepancy in the metric across different classes. For example, when a pixel is predicted to be *fence*, the possibilities of its true labels—rider, motorcycle, and bicycle—are all less than $10^{-6}$, contrasting sharply with other classes that are in the tens of thousands. We suggest this stark difference elucidates the extreme variation and irregularity in the model's predictive accuracy for different classes.

*Recall-precision analysis.* To further detailed analysis of LT, we also show the quantitative metrics of each class on ACDC-fog[3], as shown in Figure 4 (cf. Appendix). We conduct a comparison of the results between two experiments: *Source Only* (SO) and *Adapt* (where we fine-tune the source model using 100% GT labels). Firstly, as evident in all the plots of this figure, the majority classes consistently achieve exceptionally high scores across all metrics, whereas the minority classes do not consistently perform the worst. Secondly, following the adaptation process (involving the addition of supervised information to model training), the recall of most classes shows improvement, while the precision of certain minority classes experiences a decrease. This indicates that the model is less likely to miss pixels of this class (predicting it as other classes) while becoming more prone to predicting pixels of other classes as this class. This phenomenon diverges from the patterns observed in classification tasks [44] and does not align with conventional wisdom, adding complexity to the uncovering of underlying patterns.

*Model calibration.* We conduct experiments to delve into model interpretability, aiming to unearth the primary challenges associated with the uncertainty of segmentation TTA. According to

[3]The results on the other domains of dataset ACDC are presented in Figure 6-Figure 8 (cf. Appendix).

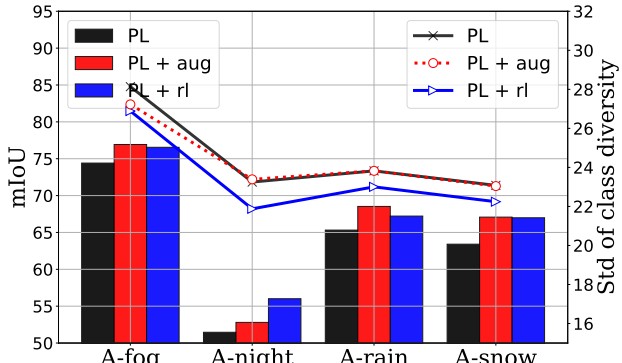

**Figure 4: Test-time augmentation and region-level training strategies can relieve LT biases. mIoU (%) and std are displayed. "PL"/"aug"/"rl" are short for pseudo-labeling/test-time augmentation/region-level, respectively.**

the results displayed in Figure 3 (a)-(d), we find that SO records the lowest ECE at 9.8%. However, TENT, TS, and SO (Aug) fail to generate improved confidence estimation after adaptation. On the other hand, TENT seems to bolster the model's performance in low confidence zones, particularly in the bins spanning from 0.1 to 0.5 as shown in Figure 3 (b). In contrast, the TS scheme exhibits subpar prediction accuracy in these low confidence bins and consistently avoids low probability predictions, as distinctly seen in Figure 3 (c). Although SO (Aug) gains the highest result (Table 2), it does not succeed in enhancing calibration. In summary, while these methods showcase their strengths in segmentation TTA, calibration remains a nuanced challenge and it is imperative to consider the interplay of various factors.

## 6.2 How to Relieve LT Biases?

Having already identified the LT problem as a key challenge in segmentation TTA, our exploration will focus on effective strategies in mitigating these biases. While re-weighting and re-sampling are prevalent methods in managing imbalanced data [51], applying these strategies at pixel-level in segmentation TTA does not yield positive results. In fact, it may lead to worse performance. As discussed in Section 4, since statistics based on pixel-level are highly unstable, we employ a re-sampling approach that focuses on

region-level. Furthermore, we also consider the test-time augmentation, which has shown to be effective in Section 5. The mIoU and the standard deviation (std) of class diversity on dataset ACDC are shown in Figure 4, displaying that both of these two strategies can relieve the LT problem. Although test-time augmentation brings improvement, its std is similar to the baseline (PL). In this way, re-sampling based on region-level demonstrates the most obvious potential.

Furthermore, we consider the individual role of augmentation and the results are displayed in Table 2 (cf. Appendix), pondering the potential of test-time augmentation to alleviate the issue of tail-class information scarcity [53]. Following this, we conduct an ablation study for test-time augmentation [26, 42] in terms of the F1 Score and mIoU. As shown in Table 3 (cf. Appendix), it is clear that employing data augmentation results in a 2.4% increase in mIoU. However, it simultaneously leads to a 0.9% decrease in the F1 Score. This suggests that the model, post-augmentation, intensifies its prediction of minority classes, leading to a simultaneous rise in both True Positive and False Positive, thereby boosting mIoU. Nonetheless, the nuanced balance of Recall and Precision in the F1 Score leads to a less pronounced change. Regarding the tail classes, we observe a notable 4.4% increase in mIoU, contrasted by a 1.1% decline in F1 Score. This showcases that while augmentation enhances the model's detection of tail classes, it does not uniformly improve its precision for these classes. In light of the above observations, we conclude that Aug partially relieves LT biases in segmentation TTA. In the future, we will explore integrating region-level segmentation and Aug to address the LT problem in segmentation TTA.

# 7 OUR PROPOSED BENCHMARK: TTAP

Prompt tuning is an inspirational technique that can produce additional textual instructions to fine tune large-scale Natural Language Processing (NLP) models for specific downstream tasks [24]. Inspired by this fact, we attempt to investigate the applicability of visual prompt tuning (VisPT) in segmentation TTA. Recently, VisPT has also been introduced into TTA methods for parameter-efficient transfer, i.e., $\mathbf{x} = \mathbf{x} + \mathcal{P}$, where $\mathcal{P}$ is the visual prompt. DePT [7] is derived from VPT [15], which introduces a small amount of task-specific learnable parameters into the input space while freezing the entire pre-trained transformer block during adaptation. DVPT [6] introduces both domain-specific and domain-agnostic prompts to prevent catastrophic forgetting and error accumulation. Compared to DVPT, SVDP [47] proposes sparse visual domain prompts to reserve more spatial information of the input image. UniVPT [27] suggests a lightweight prompt adapter to progressively encode informative knowledge into prompts, thereby improving their spatial robustness.

Based on the above analysis, we suggest that generating visual prompts can leverage image priors to provide a straightforward and effective strategy, i.e., frequency domain [41]. By combining RGB and frequency domain, we can uncover a richer set of image priors, proving invaluable for the creation of visual prompts. The comparative results displayed in Table 5 (cf. Appendix) indicate that VisPT a promising technique in segmentation TTA.

To further explore the potential of Vis**PT** in segmentation **TTA**, we propose a benchmark named **TTAP**. TTAP is based on VisPT

**Table 5: Comparisons between TTAP and other methods (mIoU, %). The computational time (minute) on dataset ACDC is also displayed. The computational time of CoTTA is more than ten times of TTAP, while our accuracy is just slightly lower than CoTTA.**

| Method | CS (GTA) | CS (Syn) | CS-fog | CS-rain | ACDC (time) | Avg. |
|---|---|---|---|---|---|---|
| SO | 68.6 | 51.1 | 74.2 | 66.6 | 56.3 (1.7) | 63.4 |
| TENT | 67.8 | 50.4 | 73.9 | 66.8 | 53.1 (2.0) | 62.4 |
| CoTTA | 65.5 | 50.4 | 75.2 | 68.7 | 57.6 (68.2) | 63.6 |
| DePT | 65.1 | 48.2 | 60.1 | 57.1 | 52.6 (5.0) | 56.6 |
| DVPT | 66.3 | 48.6 | 67.7 | 63.3 | 56.5 (5.5) | 60.5 |
| UniVPT | 60.2 | 43.3 | 60.1 | 44.2 | 36.2 (20.9) | 48.9 |
| SVDP | 69.1 | 52.2 | 67.8 | 64.3 | 57.2 (75.5) | 62.1 |
| **TTAP** (ours) | **72.1** | **57.6** | **76.0** | **71.0** | 57.2 (6.0) | **66.8** |

and our previous observations. First, we generate the visual prompt for each test sample using image priors (Section 7). Then, we adopt the TS framework to produce high-confidence pseudo-labels to refine the visual prompts. The time-consuming (Section 5) technique of Aug is not adopted, since online adaptation demands a highly time efficiency. Finally, we update the attention module and visual prompts, since it is hard to address distribution shifts solely depending on normalization layers in transformer-based architectures (Section 4). The comparative results are displayed in Table 5, where it is clear that TTAP achieves outstanding performance. Although CoTTA [42] achieves higher results on ACDC dataset, it is time-consuming due to the policy of Aug. In contrast, our proposed approach TTAP only updates limited parameters without augmentation and the computational time is less than 10% of CoTTA. Furthermore, our average performance is higher than all the other approaches.

# 8 CONCLUSIONS

In TTA community, an open question still remains to be investigated: Can classic test-time adaptation strategies be effectively applied in semantic segmentation? Our purpose is to address this question to assist both experienced researchers and newcomers in better understanding segmentation TTA. In this paper, we provide extensive experiments and comprehensive analysis to investigate the applicability of popular TTA strategies such as normalization and teacher-student scheme. Ground-truth labels are also introduced to examine how pseudo-labels affect the single-model. Experimental results indicate that those classic strategies do not perform well in segmentation TTA. Meanwhile, we also attempt to disclose the fundamental reasons and suggest some possible solutions, such as updating the attention module and integrating region-level segmentation.

Besides the regular observations, we discover that visual prompt tuning (VisPT) is a promising solution to address segmentation TTA. Further, based on VisPT and the those observations, we propose a novel benchmark named **TTAP** which has also been proved to be effective. More information such as Tables, Figures and analysis can be found in the Supplementary Material. We hope that more researchers can join the TTA community and build a common practice for segmentation.

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
