# OpenReview forum: "From Question to Exploration: Can Classic Test-Time Adaptation Strategies Be Effectively Applied in Semantic Segmentation?"
_acmmm.org/ACMMM/2024/Conference — MM2024 Poster_

### Official Review · Reviewer_UBdJ · 2024-05-14

**Rating:** 4
**Confidence:** 3

**Summary:**

The paper investigates the effectiveness of classic test-time adaptation (TTA) strategies in semantic segmentation, a more challenging task compared to classification. The authors explore existing TTA methods and their impact on segmentation TTA, highlighting observations such as the marginal performance gains of normalization updating strategies and the potential of updating the attention module in Transformers to improve segmentation TTA outcomes. The study also delves into the teacher-student scheme in segmentation TTA, showing that the TS-EMA scheme can enhance training stability and improve model performance compared to other schemes. Furthermore, discussions on potential limitations of the TS-EMA scheme reveal insights into the impact of accurate pseudo-labels on model performance and the importance of data augmentation in segmentation TTA. The paper aims to shed light on the challenges of the long-tailed class-imbalance issue in segmentation TTA and emphasizes the complexity of semantic segmentation tasks compared to classification, providing systematic studies to assist researchers in better understanding segmentation TTA. Finally，based on the above observations, TTAP method is proposed, and promising performance have achieved on several datasets.

**Strengths:**

1. The strengths of the paper lie in its comprehensive investigation of classic test-time adaptation (TTA) techniques for semantic segmentation, addressing an open question in the field.
2. The paper offers several insights by exploring the effectiveness of classic TTA strategies in the context of semantic segmentation, a more complex task compared to classification.
3. The paper's systematic studies provide valuable guidance for both experienced researchers and newcomers in understanding segmentation TTA, contributing to the advancement of the TTA community.
4. The proposed benchmark named TTAP, based on visual prompt tuning and comprehensive analysis, showcases the practical applications and effectiveness of the findings in segmentation TTA.

**Limitations:**

1. The paper's weakness lies in the lack of a clear demonstration of implementation in the proposed test-time adaptation (TTA) strategies for semantic segmentation. While the paper extensively explores classic TTA techniques and introduces the concept of visual prompt tuning (VisPT) for segmentation TTA, it falls short in explicitly showcasing how these approaches significantly advance the field beyond existing methods.
2. The proposed benchmark TTAP seems to be an integration of existing methods, so the contributions and novelty about the TTAP should be explained.
3. The absence of a comparative analysis with state-of-the-art segmentation TTA methods or a detailed discussion on how the proposed strategies address specific limitations in prior works is recommended to be discussed.
4. The paper's second contribution presents a confusing outcome wherein the teacher-student (TS) schemes generate high-quality pseudo-labels without improving the accuracy of Test-Time Adaptation (TTA).
5. The paper conducts a series of experiments to validate the efficacy of Batch Normalization (BN) updating and the teacher-student scheme. Yet, many results are relegated to supplementary materials; it is recommended that these findings be included in the main text.
6. The accuracy of pseudo labels is a critical factor for all TTA methods, and these labels are often inherently imprecise in practice. Despite this, the authors have employed ground truth labels to assess effectiveness of different schemes. It remains unclear how the insights gained from ground truth validation can be confidently extrapolated to real-world scenarios where only pseudo labels are available.

**Suitability:**

2

---

### Official Review · Reviewer_E1V8 · 2024-05-16

**Rating:** 4
**Confidence:** 2

**Summary:**

The main question of this manuscript is: Can the classic Test-Time Adaptation (TTA) strategy be effectively applied to semantic segmentation tasks? Although TTA has achieved remarkable success in classification problems, researchers may think that these classic TTA techniques can be directly applied to more challenging semantic segmentation tasks. This paper aims to investigate the applicability of existing classic TTA strategies in semantic segmentation.

**Strengths:**

1. The paper provides a comprehensive evaluation of various classic TTA strategies in semantic segmentation through extensive experiments and detailed analysis​.
2. The paper features well-structured experimental designs and exhaustive data analyses.

**Limitations:**

1. The full text lacks an overall framework that covers all discussions and methods used in the article. In addition, the proportion of abbreviations introduced in the article is unusually high, making the full text somewhat difficult to read. The author needs to further improve the readability of the paper.
2. When the article proposed the final TTAP solution, it did not introduce the relationship between prompt tuning and TTA in great detail, but we know that prompt tuning was originally used in the PEFT task. The author needs to further explain the reason for introducing prompt tuning and the specific usage method.
3. The results from table 2 shows that the effect of "PL" is less obvious than the effect of "Aug" on TS, and the method on 'single' method caused a greater performance loss. How to explain this phenomenon.

**Suitability:**

2

---

### Official Review · Reviewer_x531 · 2024-05-22

**Rating:** 4
**Confidence:** 3

**Summary:**

This paper comprehensively reviews existing TTA techniques and their performance when adapted to semantic segmentation, including normalization-based methods, the teacher-student scheme, and the longtail issue. Based on visual prompt tuning and the findings, the authors proposed a benchmark method called TTAP, which achieves compelling results with high efficiency.

**Strengths:**

1. The authors provide thorough analysis of the existing TTA techniques, and their performance when adapted to semantic segmentation, including normalization-based methods, the teacher-student scheme, and the longtail issue.
2. Several findings can shed light on the challenge of test-time adaptation for semantic segmentation.
3. The article is well-written with proper vocabulary and smooth flow.

**Limitations:**

1. The introduction of the proposed TTAP is too brief. In L887, "we generate the visual prompt for each test sample using image priors (Section 7)", I don't think Section 7 tells the reader how the visual prompts are generated.
2. What does "TS framework" in L889 refer to?
3. I believe a more detailed introduction of the proposed TTAP benchmark, including method and experimental settings will benefit the followers.

Typos:
L845 fine tune -> fine-tune
"on" is missed in "based the above findings" in L32

**Suitability:**

2

---

### Official Review · Reviewer_eg3W · 2024-05-23

**Rating:** 3
**Confidence:** 2

**Summary:**

This paper investigates the effectiveness of classic Test-time Adaptation (TTA) techniques applied to the Semantic Segmentation task. The study reveals three significant findings: First, the traditional normalization updating strategy offers minimal performance enhancements and can sometimes degrade results. Second, the teacher-student scheme does not inherently enhance performance over the original model without TTA, particularly under complex data distributions. Third, segmentation TTA is challenged by a severe long-tailed class imbalance. In response to these findings, the paper introduces a novel benchmark named TTAP, utilizing Visual Prompt Tuning (VisPT), which shows promising results in addressing these issues.

**Strengths:**

Strengths
- The paper conducts thorough evaluations of classic Test-time Adaptation (TTA) methods for classification when applied to the more complex task of semantic segmentation.
- The newly proposed TTAP method demonstrates promising results in enhancing segmentation performance.

**Limitations:**

Weakness
- The novelty of the approach is somewhat limited as vision prompting is widely used in the segmentation tasks. [i][ii]
- The paper could benefit from a more detailed explanation of the proposed TTAP method. Specific clarifications are needed on how visual prompts are generated for each test sample using image priors, as mentioned in lines 887-888. Additionally, in lines 892-893, it is unclear whether all parameters in the attention module are updated or just a subset, and the method of updating the visual prompts remains vague.
- For improved clarity and reference, it is recommended to include citations in all tables for different methods.


[i] Ye, Yiwen, Yutong Xie, Jianpeng Zhang, Ziyang Chen, and Yong Xia. "UniSeg: A Prompt-Driven Universal Segmentation Model as Well as A Strong Representation Learner." In International Conference on Medical Image Computing and Computer-Assisted Intervention, pp. 508-518. 2023.
[ii]Lüddecke, Timo, and Alexander Ecker. "Image segmentation using text and image prompts." In Proceedings of the IEEE/CVF conference on computer vision and pattern recognition, pp. 7086-7096. 2022.

**Suitability:**

2

---

### Meta-Review · Area_Chair_ucAi · 2024-06-29

**Recommendation:** Accept (Poster)
**Confidence:** 5

**Metareview:**

The paper initially got three borderline accept and one borderline reject. The authors have provided a rebuttal. After checking the rebuttal and comments of the other reviewers, the Reviewer eg3W who gave borderline reject changed the rating to borderline accept while the other reviewers kept their original rating. The paper finally got four consistent borderline accept. The AC thus thinks this paper is a borderline paper and would like to recommend it as a poster. In the final version, the authors are encouraged to include all the discussions and experiments in the rebuttal, as well as, provide more explanation for attention module and an illustration to TTAP.